# Validation of Rapid Point-of-Care Diagnostic Tests for Sexually Transmitted Infection Self-Testing Among Adolescent Girls and Young Women

**DOI:** 10.3390/diagnostics15131604

**Published:** 2025-06-25

**Authors:** Krishnaveni Reddy, Jiaying Hao, Nompumelelo Sigcu, Merusha Govindasami, Nomasonto Matswake, Busisiwe Jiane, Reolebogile Kgoa, Lindsay Kew, Nkosiphile Ndlovu, Reginah Stuurman, Hlengiwe Mposula, Jennifer Ellen Balkus, Renee Heffron, Thesla Palanee-Phillips

**Affiliations:** 1Wits RHI, Faculty of Health Sciences, University of the Witwatersrand, Johannesburg 2001, South Africa; msigcu@wrhi.ac.za (N.S.); mgovindasami@wrhi.ac.za (M.G.); nmatswake@wrhi.ac.za (N.M.); bjiane@wrhi.ac.za (B.J.); rkgoa@wrhi.ac.za (R.K.); lkew@wrhi.ac.za (L.K.); nkndlovu@wrhi.ac.za (N.N.); rstuurman@wrhi.ac.za (R.S.); hmposula@wrhi.ac.za (H.M.); tpalanee@wrhi.ac.za (T.P.-P.); 2Department of Medicine, Heersink School of Medicine, University of Alabama at Birmingham, Birmingham, AL 35294, USA; hjy1988@uab.edu (J.H.); rheffron@uabmc.edu (R.H.); 3School of Public Health, University of Washington, Seattle, WA 98195, USA; jbalkus@uw.edu

**Keywords:** vaginal swabs, self-testing options, chlamydia, gonorrhea, trichomoniasis, validation, STI, sensitivity, accuracy

## Abstract

**Background/Objectives**: High rates of sexually transmitted infections (STIs) increase HIV transmission risk among adolescent girls and young women (AGYW) in South Africa. AGYW prefer discreet self-testing options for HIV and pregnancy; however, other STI self-testing options are currently unavailable in this region. **Methods**: Seven *Chlamydia trachomatis* (CT), *Neisseria gonorrhea* (NG) and *Trichomonas vaginalis* (TV) assays were validated for AGYW self-test use (using self-collected vaginal samples) in a cross-sectional study (PROVE). Paired GeneXpert^®^ NG/CT (Cepheid_®_, Sunnyvale, CA, USA) and OSOM^®^ Trichomonas test (Sekisui Diagnostics, Burlington, MA, USA) results from nurse-collected samples served as reference results to calculate sensitivity, specificity, positive predictive values (PPV), and negative predictive values (NPV). One test, the polymerase chain reaction (PCR)-based Visby Medical™ Sexual Health Test device (Visby Medical™, San Jose, CA, USA), was validated for accuracy of positive test results using self-collected samples and home-based testing in a longitudinal follow-up study enrolling AGYW aged 16–18 years. Paired GeneXpert^®^ NG/CT and TV results from nurse-collected vaginal samples served as reference tests. **Results**: In PROVE, 146 AGYW contributed 558 paired samples. The Visby Medical™ Sexual Health Test exhibited moderate to high sensitivity (66.7–100%), specificity (80–100%), NPV (66.7–100%), and PPV (66.7–100%) for NG, CT, and TV. The remaining tests’ performances were markedly lower. In the longitudinal study, 28 AGYW contributed 84 paired samples, and the Visby Medical™ Sexual Health Test demonstrated 100% accuracy of positive results for CT, NG, and TV. **Conclusions**: The Visby Medical™ Sexual Health Test demonstrated high reliability as a potential option for AGYW to discreetly self-test for multiple STIs concurrently. Testing of its acceptability, utility, and feasibility in a larger sample of AGYW is in progress.

## 1. Introduction

With more than a million infections acquired daily among people aged 15–49 years globally, sexually transmitted infections (STIs) caused by *Chlamydia trachomatis* (CT), *Neisseria gonorrhea* (NG), and *Trichomonas vaginalis* (TV), are among the most common infections worldwide [1,2]. South Africa has some of the highest rates of STIs globally, with modeled NG and CT prevalence estimates of 6.6% and 14.7% for women and 3.4% and 6.0% for men [3,4]. Adolescent girls and young women (AGYW) in particular are disproportionately impacted compared to their male peers due to a combination of factors, including biological, socioeconomic, religious, and cultural factors [5,6]. Recently published South African data indicates rates of 62.8% for any STI (55.9% CT, 15.2% NG, and 9.5% TV) and 42.5% (33.4% CT, 7.9% NG, and 7.1% TV) among adolescent girls aged 14–19 years in Western Cape and KwaZulu Natal, respectively [7] and 36–37.3% (28–30.9% CT, 3.7–7% NG, and 1–6.2% TV) among AGYW aged 16–25 in Johannesburg [8,9]. When left untreated, sequelae include acute urogenital conditions (i.e., cervicitis, urethritis, vaginitis, genital ulceration, pelvic inflammatory disease, ectopic pregnancy, infertility, chronic pelvic pain), arthritis, and poor birth outcomes [1,10,11]. Additionally, these common STIs have been implicated in increasing the risk of HIV acquisition and transmission [12].

STI management programs in South Africa generally rely on syndromic management [4,13]. While this approach provides a less expensive and practical approach to STI control, its effectiveness is limited to individuals who experience and disclose symptoms. The prevalence of asymptomatic STIs is, however, very high among AGYW, with studies in South Africa demonstrating up to 78% asymptomatic STIs [14,15]. This would be missed by syndromic management. Additionally, syndromic management relies heavily on the clinical judgment of healthcare providers and has low specificity, resulting in the use of broad-spectrum, empiric treatments repeatedly, which have subsequently contributed to the emergence of antimicrobial resistance [16,17]. Therefore, targeted approaches for identifying STIs regardless of symptoms are needed to effectively diagnose, treat, and ultimately reduce the burden of STIs among those exposed.

One sought-after strategy is point-of-care (POC) diagnostic testing, and several platforms exist (including Cepheid_®_ GeneXpert^®^, a real-time polymerase chain reaction (PCR)-based nucleic acid amplification test used to detect CT, NG, and TV in laboratory settings). Per the World Health Organization’s (WHO) “ASSURED” criteria, however, these tests should be affordable, sensitive, specific, user-friendly, rapid and robust, equipment-free, and deliverable to end users [18], and not many meet all of the stipulated criteria. The option of self-testing using POC rapid test kits coupled with self-sampling has seen tremendous success for HIV [19], pregnancy [20], and SARS-CoV-2 [21]. POC self-testing for STIs has the potential to increase access to and frequency of screening, subsequently shortening the time to awareness of infection, desire to seek treatment, treatment initiation, and prevention of onward transmission [22,23]. We conducted a full landscape review of diagnostic test kits for CT, NG, and TV and narrowed these down using our criteria of being rapid, easy to use, not requiring a reader, and available in South Africa. Through this process, we identified six antigen-based lateral flow diagnostic test kits for CT, NG, and/or TV that could be candidates for STI self-testing. We later added one fully integrated, automated PCR in vitro diagnostic test that had recently been granted a Clinical Laboratory Improvement Amendments of 1988 (CLIA) waiver from the U.S. Food and Drug Administration (FDA) (i.e., defined as a simple test with a low risk for an incorrect result). We systematically compared the test characteristics of these STI kits, leveraging two longitudinal cohorts of AGYW in South Africa.

## 2. Materials and Methods

### 2.1. Initial Cross-Sectional Validation—PROVE Study

Sexually active, cis-gender, STI positive AGYW aged 18–25 years, screening for or already enrolled into an existing longitudinal follow-up study on acceptability of POC STI testing and expedited partner therapy (ARISE, ct.gov #: NCT06570733) at the Wits RHI Research Centre clinical research site (RC CRS) in Johannesburg, South Africa and who were willing to provide additional samples, were invited to co-enroll in PROVE, a cross-sectional study to validate selected POC rapid STI diagnostic tests (ct.gov #NCT06566677). Exclusion criteria included having any condition that, in the opinion of the study investigators, could interfere with completion of study procedures or make study participation unsafe. These AGYW were identified through recruitment drives at local public sector health facilities and spaces frequented by AGYW, word-of-mouth referrals, and peer-to-peer recruitment.

PROVE study procedures, included informed consent and collection of four self-collected vaginal swab samples, were completed after laboratory-based GeneXpert^®^ CT/NG (Cepheid_®_, Sunnyvale, CA, USA) and OSOM^®^ TV (Sekisui Diagnostics, Burlington, MA, USA) testing were conducted as part of the ARISE study’s standard enrolment and follow up study procedures. One to three sterile Dacron swabs were used to collect samples for the lateral flow diagnostic tests (one swab for TV testing and three swabs each for CT and NG testing) and one Visby Medical™ Sexual Health Collection Kit (including a swab and collection tube) (Visby Medical™, San Jose, CA, USA) for the Visby Medical™ Sexual Health Testing (Visby Medical™, San Jose, CA, USA) (Table 1). Once self-collected, these validation samples were transported to the on-site laboratory and stored at 2–8 °C until testing by trained laboratory staff using the identified diagnostic kits and accompanying test-specific package insert instructions (within 24 h).

This process allowed the generation of paired sample results for the validation of the identified diagnostic STI test kits against GeneXpert^®^ CT/NG and OSOM^®^ TV tests as laboratory reference tests (Figure 1).

AGYW who tested positive for STIs per GeneXpert^®^ CT/NG and OSOM^®^ TV were treated with directly observed therapy (after collection of additional validation samples) as part of the ARISE study STI management at the study site and according to South African National Department of Health (SA NDoH) and WHO STI Treatment Guidelines [13,31]. These participants were also offered STI treatment to deliver to their partner/s (e.g., expedited partner therapy) or partner notification slips to support partner treatment.

### 2.2. Longitudinal Cohort with Observed New STI Diagnoses and Visby Medical™ Sexual Health Test Validation—PALESA Study

Separately, literate, sexually active, non-pregnant, HIV negative, cisgender AGYW aged 16–18 years, who discontinued oral HIV pre-exposure prophylaxis (PrEP) medication within the past 6 months, were enrolled into a pilot study investigating the feasibility of conducting a powered randomized controlled trial of STI self-testing for AGYW (PALESA, ct.gov #NCT06030856). Exclusion criteria included being unwilling to comply with study procedures, already participating in a research study involving drugs, medical devices, or vaccines for STI prevention or treatment, or having any condition that, in the opinion of the study investigators, could interfere with completion of study procedures or make study participation unsafe. This study was implemented at the Wits RHI RC CRS, and recruitment strategies included outreach to public clinics and spaces frequented by AGYW as well as peer-to-peer recruitment strategies.

The Visby Medical™ Sexual Health Test performed best in the initial cross-sectional validation (see Results), thus was selected for further validation in this follow-on study. The study visit schedule included monthly visits for 6 months and was a combination of in-person visits (screening and enrolment, month 3, month 6) and virtual visits (months 1, 2, 4, 5). At the screening and enrolment visit, AGYW were randomized in a 1:1 ratio to receive: (1) self-administered behavioral risk assessment plus Visby Medical™ Sexual Health Test kits for self-test use in the virtual visit months or (2) self-administered behavioral risk assessment alone. Following randomization, participants in the self-testing arm received in-person instruction along with Visby Medical™ reference material for use of the Visby Medical™ Sexual Health kits (Sexual Health Quick Reference Guide [32]) (and Vaginal Specimen Collection Kit (Self-Collection Instructions [33]). They were also provided with a link to an instructional video: https://www.youtube.com/watch?v=9SpFpfGrV2g (accessed on 28 June 2023) and participated in a clinical staff-supervised, in-person practice run for self-sampling and self-testing. Results were verified against GeneXpert^®^ CT/NG and/or GeneXpert^®^ TV testing conducted as part of this visit. These participants then received two additional Visby Medical™ Sexual Health Test kits (and two Visby Medical™ Sexual Health Collection Kits including a swab and collection tube) to be used at their month 1 and month 2 virtual visits (kits for month 4 and month 5 virtual visits were dispensed at their month 3 in-person visit). When participants had a reactive test result through self-testing with the Visby kit during study follow-up, they were asked to return to the study clinic for confirmatory testing (using GeneXpert^®^ CT/NG and/or GeneXpert^®^ TV (Cepheid_®_, Sunnyvale, CA, USA)). This process allowed the generation of paired sample results to check the accuracy of Visby Medical™ Sexual Health Test positive results against GeneXpert^®^ CT/NG and TV tests as laboratory reference tests. All participants who were confirmed STI positive were treated according to SA NDoH guidelines with directly observed therapy and offered partner notification slips to support partner treatment at local clinics.

### 2.3. Sample Self-Collection Procedures

There were no restrictions on sample collection apart from self-sampling during menstruation, just after sex, and within 48 h of using vaginal products or antiperspirants/deodorants in the genital area to avoid substance interference in the testing process. Participants were instructed to hold the swab between their thumb and forefinger, carefully insert the soft tip of the swab 5 cm into the vagina opening and to gently rotate it eight times (for 10–30 s), making sure that the swab touched the vagina walls. Clinical staff provided guidance and were available to assist during in-clinic visits.

### 2.4. Data Collection

Data for PROVE and PALESA participants were collected using REDCap, a secure web application for building and managing online surveys and databases. Additionally, for participants in the STI self-testing arm of PALESA, self-testing results, including a photo of the completed test kit, were uploaded into REDCap.

### 2.5. Statistical Analysis

For the PROVE cross-sectional study, we used results from the matching laboratory-based reference tests in the ARISE longitudinal follow-up study to calculate sensitivity, specificity, positive predictive value (PPV), and negative predictive value (NPV) of the selected seven STI assays (with 95% confidence intervals (CIs)) to check stability and correctness of each diagnostic test kit. Kappa Coefficients and likelihood ratios with 95% CIs were also calculated to evaluate the clinical performance of test kits. We considered Kappa values ≤ 0 as indicating no agreement and 0.01–0.20 as none to slight, 0.21–0.40 as fair, 0.41–0.60 as moderate, 0.61–0.80 as substantial, and 0.81–1.00 as almost perfect agreement [34]. Bar charts were generated to visually present the statistics.

For the PALESA longitudinal study, negative results from the Visby Medical™ Sexual Health Tests were not confirmed, and we were limited to presenting accuracy information on the positive results only. This study was not powered for statistical comparisons since it was meant to determine the feasibility of conducting a powered randomized controlled trial. All analysis was performed in SAS 9.4.

### 2.6. Ethics

The ARISE, PROVE, and PALESA protocols were approved by the Wits Human Research Ethics Committee (HREC) (#210614, #220711, #220909). All participants aged ≥18 years provided written informed consent. Those aged 16–17 years provided assent alongside written consent from their parent/legal guardian.

## 3. Results

### 3.1. Study Participants’ Socio-Demographic Characteristics

In the PROVE cross-sectional validation study, 146 participants (median age (IQR) 21 (19–23)) contributed towards 558 paired samples (Figure 1). For the PALESA longitudinal cohort, 28 participants (median age (IQR) 18 (17–18)) were randomized to use the Visby Medical™ Sexual Health Test as a self-test and contributed 84 paired samples. The cohorts were similar in terms of their distribution of race (majority black) and religion (majority Christian) (Table 2).

### 3.2. Initial Cross-Sectional Validation Results—PROVE Study

Results from self-collected vaginal swab samples run using four rapid tests to detect CT (Cromatest^®^ Linear Chlamydia Cassette, NADAL^®^ Chlamydia Test, JD Biotech *C. trachomatis*/*N. gonorrhea* Ag Combo Rapid Test, and Visby Medical™ Sexual Health Test) were compared to CEPHEID GeneXpert^®^ CT/NG results. Of these, the Visby Medical™ Sexual Health Test exhibited the highest sensitivity at 78.8% along with 80.0% specificity, 87.5% PPV, and 66.7% NPV. For the remaining three tests, sensitivity varied from 0 to 9.7%, specificity was 100%, PPV was 100% (except for the JD Biotech *C. trachomatis*/*N. gonorrhea* Ag Combo Rapid Test), and NPV ranged from 55.6% to 58.7%. A moderate agreement was observed with the Visby Medical™ Sexual Health Test (k = 0.55) (Table 3 and Figure 2a).

Results from self-collected vaginal swab samples run using four rapid tests to detect NG (JD Biotech *C. trachomatis*/*N. gonorrhea* Ag Combo Rapid Test, ACRO Biotech Inc Gonorrhea Rapid Test Cassette, Rapid Labs Gonorrhea Rapid Test Device and Visby Medical™ Sexual Health Test) were compared to CEPHEID GeneXpert^®^ CT/NG results. 

The Visby Medical™ Sexual Health Test exhibited 100% sensitivity, specificity, PPV, and NPV. For the remaining three tests, sensitivity varied from 0 to 33.3%, specificity was 100%, positive predictive value was 100% (except for the JD Biotech *C. trachomatis*/*N. gonorrhea* Ag Combo Rapid Test), and NPV varied from 56.5% to 68.4%. A perfect agreement was observed with the Visby Medical™ Sexual Health Test (k = 1.00) (Table 3 and Figure 2b).

Results from self-collected vaginal swab samples run using two rapid tests to detect TV (JD Biotech *Trichomonas vaginalis* Antigen Rapid Test Kit and Visby Medical™ Sexual Health Test) were compared to the OSOM^®^ TV results. The Visby Medical™ Sexual Health Test exhibited 66.7% sensitivity, 90.9% specificity, 66.7% positive predictive value, 90.9% negative predictive value, and moderate agreement (k = 0.58). The JD Biotech *Trichomonas vaginalis* Antigen Rapid Test Kit exhibited comparatively lower sensitivity at 64.3%, 100% specificity, 100% PPV, 79.2% NPV, and substantial agreement (k = 0.67) (Table 3 and Figure 2c).

### 3.3. Longitudinal Cohort Results—PALESA Study

Relative to nurse-collected samples tested with laboratory-based CEPHEID GeneXpert^®^ testing, the Visby Medical™ Sexual Health Test was accurate on positive results 100% for CT, NG, and TV when paired with participant self-collection of vaginal samples and self-testing (Table 4).

## 4. Discussion

In the PROVE cross-sectional validation, the Visby Medical™ Sexual Health test exhibited moderate to high sensitivity, specificity, PPV, and NPV for CT, NG, and TV, as well as moderate to perfect agreement. This drove its selection for our PALESA self-test validation with AGYW, followed longitudinally for 6 months. We then observed 100% accuracy for this test to identify positive results when it was used by AGYW as a self-test with self-sample collection. We were unable to calculate specificity and predictive values within the PALESA longitudinal study due to our study design, which did not incorporate confirmation of negative test results. High ranges of sensitivity (CT: 93·9%, NG: 100%, TV: 100%) and specificity (CT: 82·6%, NG: 95·4%, TV: 77·6%) for the Visby Medical™ Sexual Health test when used with self-collected samples (and performed onsite by a laboratory technician) were recently reported within an AGYW cohort study implemented at the Wits RHI Research Centre CRS [35]. High (>90%) sensitivity is desirable in an STI self-screening test to avoid unnecessary anxiety, inconvenience, discomfort, relationship dissolution (due to result uncertainty), and medication use for the individual as well as costs associated with time and personnel for health services [36] that may come from false positive results. High specificity is also desirable to ensure that those with infections seek adequate and timely treatment for themselves and their sexual partners to reduce the persistence of STIs. Recently, a new version of the Visby Medical™ Sexual Health test, the Visby Medical™ Women’s Sexual Health Test, received FDA approval as an over-the-counter test, demonstrating concurrence in the field with our conclusions that this test is viable for self-use [37]. The Visby Medical™ Women’s Sexual Health Test works similarly to its predecessor but communicates securely to a Visby Medical™ App and displays results when the test is complete.

The Visby Medical™ Sexual Health test performed well in its detection of true positive and negative cases in the cross-sectional validation and true positive cases in the longitudinal follow-up validation. With further practice on vaginal swab self-sampling and self-testing using the Visby Medical™ Sexual Health test, we likely will see better concordance with the gold standard Cepheid GeneXpert^®^ platform. The Visby Medical™ Sexual Health test offers a range of benefits when compared to GeneXpert^®^. GeneXpert^®^ testing requires the set-up of the GeneXpert^®^ instrument, regular calibration/servicing, and skilled staff to operate it. As a result, it has a high cost through commercial laboratories (~USD 50 for GeneXpert^®^ TV + ~USD 57 for GeneXpert^®^ CT/NG). Testing for NG, CT, and TV through this platform is also more burdensome on the patient as it requires two swabs (including endocervical swab collection, which requires a healthcare provider and speculum use) as well as a 90-min wait time for results. The Visby Medical™ Sexual Health Test, on the other hand, is U.S. FDA and CLIA-waived and has several other characteristics that further lend to its compatibility for self-sample collection and self-testing. It is discreet, requires no calibration or separate platform instrument or reader, is ready to use immediately when connected to a power source, and requires only a single vaginal swab to test for NG, CT, and TV via PCR (which is more sensitive then antigen-based tests). The main barriers to its use are local availability and cost (~USD 60 per kit). For the purposes of this work, Visby Medical™ Sexual Health Test kits were imported specifically for research purposes by applying to the Section 21 unit of the South African Health Products Regulatory Authority (SAHPRA) which is responsible for processing and evaluating applications for access to unregistered medicines and devices within South Africa. Other barriers include that it is not a reusable test device and that its housing is currently made of plastic, which has a significant environmental carbon footprint. While integrating the Visby Medical™ Sexual Health Test into public and personal healthcare may initially be at a higher cost, this upfront outlay may potentially be worth the effort considering economies of scale related to the reduced use of antimicrobials, burden on the healthcare system, and persistent rates of STI transmission. More research to explore the cost-benefit calculus of this kit or similar is warranted.

By using a self-testing STI kit, AGYW would be able to rapidly self-test for STIs in convenient and private locations, which has the potential to address often reported issues of discomfort, stigma, victimization, lack of privacy in public healthcare facilities, and transportation challenges. Should an AGYW test positive for any of the STIs (CT/NG/TV), they would then be able to contact a healthcare facility to secure treatment. This process could further be streamlined through a web-based application linked to a designated clinic, allowing users to upload a photo of their test result and notify a healthcare provider, who could schedule them for treatment. This would result in timely management of STIs, reduced persistence, and reduced waiting time in burdened healthcare facilities. This would also reduce reliance on syndromic management and improve precision in antibiotic prescription, potentially contributing to reductions in antimicrobial resistance.

In terms of validation limitations, the poor outcomes observed with the lateral flow antigen-based kits in the PROVE cross-sectional validation study could be due to the characteristics of the test kit itself, sample type, sample adequacy, or the self-collection process. We used self-collected vaginal swabs to validate the selected test kits for self-test use due to our desire to identify a test that could be fully user-controlled. However, only the JD Biotech *C. Trachomatis*/*N. gonorrhea* Ag Combo Rapid Test, JD Biotech *Trichomonas vaginalis* Antigen Rapid Test Kit, and the Visby Medical™ Sexual Health Test included vaginal swabs as a recommended sample type (Table 1). The recommended sample type for the remaining tests (Cromatest^®^ Linear Chlamydia Cassette, NADAL^®^ Chlamydia Test, Rapid Labs Gonorrhea Rapid Test Device, and ACRO Biotech Inc. Gonorrhea Rapid Test Cassette) was female cervical swabs, which are more challenging for a woman to self-collect. The samples may have also been inadequate as self-collecting vaginal swab samples were new to participants and only collected after the nurse-collected samples for use on the GeneXpert^®^ CT/NG and OSOM™ TV platforms had already been collected. Additionally, validation using the self-collected samples was not done immediately upon collection (due to the time of sample collection during the study visit), but was stored at 2–8 °C and tested within 24 h. Although this was permitted per package insert, we cannot be certain that this did not impact validation results. Further validation work with these lateral flow assays using the recommended sample types (which include male urine as well as urethral and glans specimens) would be valuable to assess their performance for STI screening among women and men.

With regard to the longitudinal study of home collection and self-testing, the PALESA study was small, designed to demonstrate proof-of-concept that commercially available rapid STI test kits could be used by AGYW discreetly in their homes or other locations. The few false positive results observed may potentially be attributed to storage temperatures that varied in AGYW’s homes where temperature regulation, like air conditioning and heating, is lacking as many came from informal housing, inconsistent electricity supply due to loadshedding schedules [38], or inadequate sampling and testing procedures. Visby Medical™ Sexual Health Test kits were provided to participants to take home with them and store until the time of their next required use (in 1–2 months) along with instructions about the required storage conditions (2–30 °C), however, Johannesburg experienced several heatwaves between October 2023 and February 2024 [39] (during the conduct of the study) and some of the test kits were likely exposed to higher temperatures (up to 35 °C) during this time. While some participants may have had access to refrigerators, it is unlikely they felt comfortable storing their test kits there due to disclosure concerns. For future work, alternative ways to deliver kits to AGYW will be explored instead of issuing them multiple kits. Additionally, while participants were provided with a specific international plug adapter, the kit requires an uninterrupted source of electricity during its run. Many participants experienced electricity shortages due to outages or loadshedding—a widespread and ongoing reduction in electricity supply common in South Africa—as well as poor service delivery. Despite training provided at enrolment and when needed, sample collection and testing may have been performed sub-optimally in the participants’ homes. For instance, the test should not be used if antiperspirants and deodorants or vaginal products (including douches, washes, lubricants, vaginal wipes, vaginal moisturizers, or feminine hygiene spray) were used in the genital area within 48 h of sample collection. These practices are common among South African women and are often motivated by hygiene, enhancing sexual pleasure, and ensuring fidelity [40].

This work represents one of the first evaluations of STI self-testing, using point-of-care diagnostic STI assays (designed for laboratory or healthcare provider use) and self-collected vaginal samples, among AGYW in South Africa, a population at disproportionately high risk for both STIs and HIV. Key strengths include the evaluation of the Visby Medical™ Sexual Health Test’s performance (and self-sampling) in both clinic-based and home-based (unsupervised, real-world) settings and the use of well-established reference standards (GeneXpert^®^ for NG/CT and TV and OSOM^®^ for TV) to add methodological rigor to our validation process.

## 5. Conclusions

Our validations provide positive preliminary data to support further research on the acceptability, feasibility, and accuracy of STI self-screening using the Visby Medical™ Sexual Health Test (and its recently FDA-approved counterpart, the Visby Medical™ Women’s Sexual Health Test) with a larger cohort of AGYW. This work holds significant potential to increase access to STI screening for AGYW (and possibly other population groups although the Visby Medical™ Sexual Health Test’s performance has not yet been established with other specimen types apart from vaginal swabs), offering the benefits of privacy and clearer diagnostic outcomes as well as reduced turn-around times to targeted treatment initiation. There is substantial value in the subsequent reduction of STI sequelae among this key population as well as the downstream prevention of onward transmission and potentially reduced antibiotic resistance development.

## Figures and Tables

**Figure 1 diagnostics-15-01604-f001:**
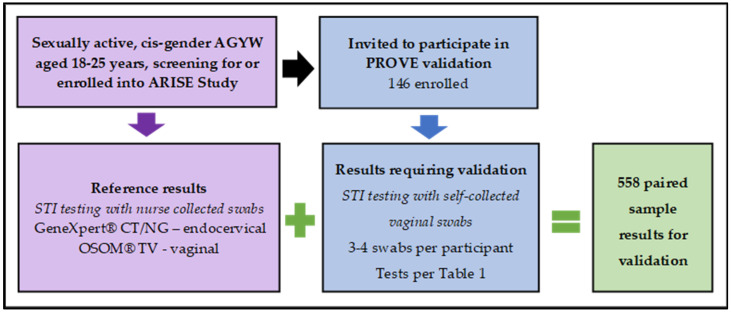
Outline of initial cross-sectional validation process.

**Figure 2 diagnostics-15-01604-f002:**
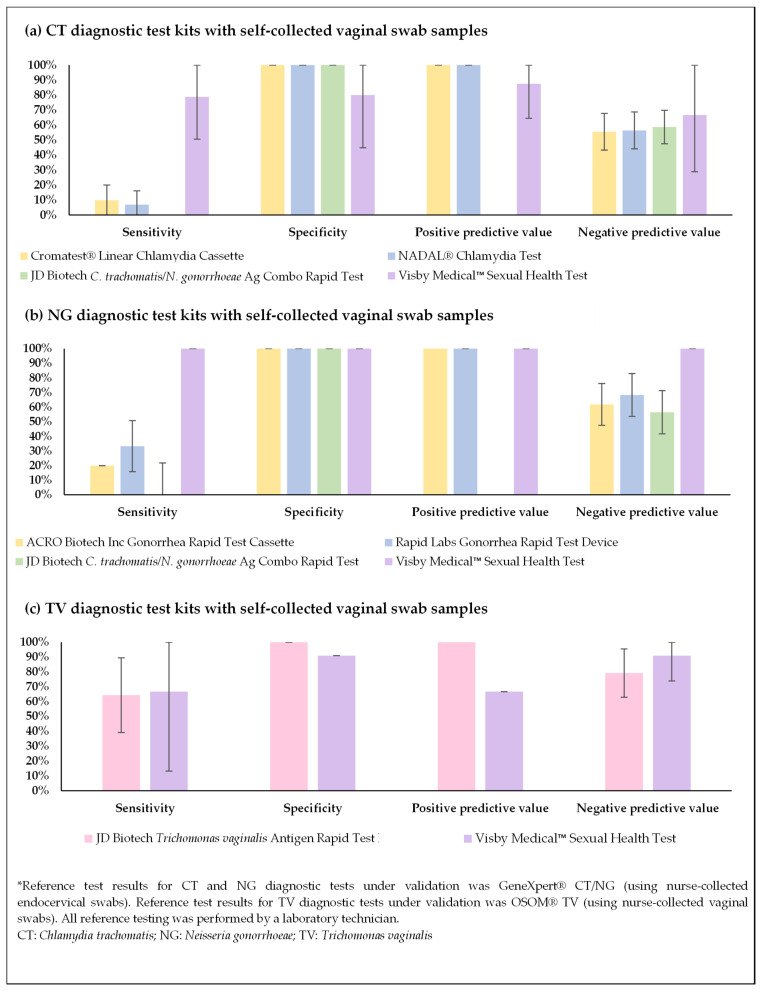
Graphical representation of PROVE cross-sectional validation test characteristics by STI.

**Table 1 diagnostics-15-01604-t001:** Diagnostic test kits selected for validation and recommended sample types (feminine), comparison tests, and associated sensitivity and specificity per package insert.

Test	Recommended Sample Type	Comparison Test	Sensitivity	Specificity
Cromatest^®^ Linear Chlamydia Cassette (Linear Chemicals S.L., Barcelona, Spain) [24]	Cervical swab	PCR	90.2% (76.9–96.5%) *	96.0% (91.2–99.4%) *
2. NADAL^®^ Chlamydia Test (nal von minden GmbH, Moers, Germany) [25]	Cervical swab	PCR	90.2% (76.9–96.5%) *	96.0% (91.2–99.4%) *
3. JD Biotech *C. trachomatis*/*N. gonorrhea* Ag Combo Rapid Test (JD Biotech^®+^, Jinan, China) [26]	Urine/vaginal swab	Clinical diagnostics	CT 87% NG 93%	CT 88% NG 90%
4. Rapid Labs Gonorrhea Rapid Test Device (Rapid Labs Ltd., Colchester, UK) [27]	Cervical swab	Culture	94.4% (86.2–98.4%) *	96.9% (91.3–99.4%) *
5. ACRO Biotech Gonorrhoea Rapid Test Cassette (ACRO Biotech, Inc., Rancho Cucamonga, CA, USA) [28]	Cervical swab	Culture	94.4% (86.2–98.4%) *	96.9% (91.3–99.4%) *
6. JD Biotech *Trichomonas vaginalis* Antigen Rapid Test Kit (JD Biotech^®+^, Jinan, China) [29]	Urine/vaginal swab	Speculum exams	100%	99%
7. Visby Medical™ Sexual Health Test [30]	Vaginal swab	NAAT	CT 97.4% (93.5–99.0%) * PPA NG 97.8% (88.4–99.6%) * PPA TV 99.3% (96.0–99.9%) * SN	CT 97.8% (96.9–98.4%) * NPA NG 99.1% (98.5–99.4%) * NPA TV 96.7% (95.8–97.5%) * SP

* 95% Confidence Interval; PCR: polymerase chain reaction; CT: *Chlamydia trachomatis*; NG: *Neisseria gonorrhea*; TV: *Trichomonas vaginalis*; NAAT: Nucleic Acid Amplification Tests; PPA: positive percent agreement; NPA: negative percent agreement; SN: Sensitivity; SP: Specificity.

**Table 2 diagnostics-15-01604-t002:** Study participants’ socio-demographic characteristics.

Demographic Characteristic		Participants for the Initial Cross-Sectional Validation (PROVE) * (*n* = 146)	Participants for the Longitudinal Cohort (PALESA) (*n* = 28)
Age, years	Median (IQR)	21 (19–23)	18 (17–18)
Race, *n* (%)	Black	141 (96.6%)	28 (100%)
	Mixed race	1 (0.7%)	0
	Missing ^1^	4 (2.7%)	-
Religion, *n* (%)	Agnostic or no religion	8 (5.5%)	5 (17.9%)
	Christian	119 (81.5%)	23 (82.1%)
	Muslim	1 (0.7%)	-
	Traditional African beliefs	11 (7.5%)	-
	Other (Shembe, Sotho, Tsonga)	3 (2.1%)	-
	Missing ^1^	4 (2.7%)	-
Highest level of education achieved, *n* (%)	Secondary school, not complete	33 (22.6%)	20 (71.4%)
	Secondary school, complete	83 (56.8%)	7 (25%)
	College or university, not complete	20 (13.7%)	1 (3.8%)
	College or university, complete	6 (4.1%)	0
	Not collected ^2^	4 (2.7%)	-
Attending any form of school now, *n* (%) ^2^	Yes	51 (34.9%)	-
	No	91 (62.3%)	-
	Not collected ^2^	4 (2.7%)	-

* PROVE participant data collected as part of the affiliated longitudinal study (ARISE). ^1^ No demographic information as participants were screened out of the affiliated longitudinal study (ARISE). ^2^ Data not collected in PALESA.

**Table 3 diagnostics-15-01604-t003:** PROVE cross-sectional validation test characteristics by STI and diagnostic test kit (using self-collected vaginal swabs) relative to laboratory-based reference tests *.

STI	Test Kit (Number of Samples)	True Pos	False Pos	True Neg	False Neg	Sensitivity (95% CI)	Specificity (95% CI)	PPV (95% CI)	NPV (95% CI)	Kappa (95% CI)	PLR (95% CI)	NLR (95% CI)
CT *	Cromatest^®^ Linear Chlamydia Cassette (*n* = 66)	3	0	35	28	9.7% (0–20.1%)	100% (100–100%)	100% (100–100%)	55.6% (43.3–67.8%)	0.10 (−0.01–0.21)	NA	0.90 (0.80–1.01)
	2. NADAL^®^ Chlamydia Test (*n* = 64)	2	0	35	27	6.9% (0–16.1%)	100% (100–100%)	100% (100–100%)	56.5% (44.1–68.8%)	0.07 (−0.03–0.18)	NA	0.93 (0.84–1.03)
	3. JD Biotech *C. trachomatis*/*N. gonorrhea* Ag Combo Rapid Test (*n* = 75)	0	0	44	31	0%	100% (100–100%)	NA	58.7% (47.5–69.8%)	NA	NA	1.00 (1.00–1.00)
	4. Visby Medical™ Sexual Health Test (*n* = 14)	7	1	4	2	78.8% (50.6–100%)	80.0% (44.9–100%)	87.5% (64.6–100%)	66.7% (28.9–100%)	0.55 (0.11–0.99)	3.89 (0.65–23.23)	0.28 (0.08–1.02)
NG *	JD Biotech *C. trachomatis*/*N. gonorrhea* Ag Combo Rapid Test (*n* = 46)	0	0	26	20	0%	100% (100–100%)	NA	56.5% (42.2–70.9%)	NA	NA	1.00 (1.00–1.00)
	2. ACRO Biotech Inc. Gonorrhea Rapid Test Cassette (*n* = 46)	4	0	26	16	20.0% (2.5–37.5%)	100% (100–100%)	100% (100–100%)	61.9% (47.2–76.6%)	0.22 (0.03–0.41)	NA	0.80 (0.64–1.00)
	3. Rapid Labs Gonorrhea Rapid Test Device (*n* = 44)	6	0	26	12	33.3% (11.6–55.1%)	100% (100–100%)	100% (100–100%)	68.4% (53.6–83.2%)	0.37 (0.14–0.61)	NA	0.67 (0.48–0.92)
	4. Visby Medical™ Sexual Health Test (*n* = 14)	5	0	9	0	100% (100–100%)	100% (100–100%)	100% (100–100%)	100% (100–100%)	1.00 (1.00–1.00)	NA	0
TV *	JD Biotech *Trichomonas vaginalis* Antigen Rapid Test Kit (*n* = 33)	9	0	19	5	64.3% (39.2–89.4%)	100% (100–100%)	100% (100–100%)	79.2% (62.9–95.4%)	0.67 (0.43–0.92)	NA	0.36 (0.18–0.72)
	2. Visby Medical™ Sexual Health Test (*n* = 14)	2	1	10	1	66.7% (13.3–100%)	90.9% (73.9–100%)	66.7% (13.3–100%)	90.9% (73.9–100%)	0.58 (0.05–1.00)	7.33 (0.96–56.00)	0.37 (0.07–1.84)

* Reference test results for CT and NG diagnostic tests under validation were CEPHEID GeneXpert^®^ CT/NG (using nurse-collected endocervical swabs). Reference test results for TV diagnostic tests under validation was OSOM^®^ TV (using nurse-collected vaginal swabs). All reference testing was performed by a laboratory technician. CT: *Chlamydia trachomatis*; NG: *Neisseria gonorrhea*; TV: *Trichomonas vaginali*; PPV: Positive Predictive Value; NPV: Negative Predictive Value; PLR: Positive Likelihood Ratio; NLR: Negative Likelihood Ratio; NA: Not variable.

**Table 4 diagnostics-15-01604-t004:** PALESA longitudinal validation test characteristics for the Visby Medical™ Sexual Health Test kit (positive results only) relative to laboratory reference tests.

STI	Positive Visby Medical™ Sexual Health Test	Positive GeneXpert^®^ CT/NG	Positive GeneXpert^®^ TV	Accuracy of Positive Visby Medical™ Sexual Health Test Results (95% CI)
CT	9	8	-	100% (100–100%)
NG	6	4	-	100% (100–100%)
TV	2	-	1	100% (100–100%)

CT: *Chlamydia trachomatis*; NG: *Neisseria gonorrhea*; TV: *Trichomonas vaginalis*.

## Data Availability

The raw data supporting the conclusions of this article will be made available by the corresponding author on request (kreddy@wrhi.ac.za).

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
