# Peer review of "Validation of Rapid Point-of-Care Diagnostic Tests for Sexually Transmitted Infection Self-Testing Among Adolescent Girls and Young Women"

_diagnostics, 2025, doi:10.3390/diagnostics15131604_

Round 1

Reviewer 1 Report

Comments and Suggestions for Authors line 46-48 – It seems as if there are double spaces present. Check for double spaces, and remove if there are any: between words "daily" and "among", "TV," and "are among". Check the entire text and remove if they are present; line 48 – Place a closing bracket ")" after "TV"; lines 49-51 – WHO [3] on page 15 noted that there are 357 million new cases of bacterial STI in total, with a leading 142 millions in WHO Western Pacific Region and only 63 millions in WHO African Region. Firstly, WHO African Region is not equal to Southern Africa (as you said in line 49). Secondly, 63 divided by 357 is not equal to 40% (as you said in line 50). Or did you add viral STI cases to that number as well? But why, if you're writing an article about bacterial STI tests? line 129-132 – Using different comparison methods to determine the sensitivity and specificity of various methods is a questionable methodological approach. Clearly explain why different comparison methods are used for different methods in Table 1, and why sensitivity and specificity values are then compared. line 130 – Remove the misprint ";," after "polymerase chain reaction"; line 130 and 256 – There is no need for you to explain the abbreviations NG, CT and TV if you have done it before; 185-193 In the study, you calculate confidence intervals, plot them on a bar chart and draw conclusions. All statistical calculations must be described explicitly in the statistics section line 256 (Figure 2) – Write in italics "C. trachomatis" and "N. gonorrhoeae"; lines 422-492 –  It seems that the list of references was generated automatically and contains formatting errors. Please pay attention to the formatting of authors' names, volume/page numbers and online sources in accordance with the guide https://www.mdpi.com/authors/references; line 451 – hyperlink https://www.who.int/reproductivehealth/topics/rtis/pocts/en/ doesn't work (This page cannot be found);. line 460 – hyperlink https://www.cdc.gov/hiv/policies/data/self-testing-issue-brief.html#print doesn't work (The page you're looking for was not found);

Reviewer 2 Report

Comments and Suggestions for Authors

Thank you for the opportunity to review the manuscript.

It is worth publishing after editing.

Please find below my comments:

  1. The title is too long. Suggest reshaping it. Suggested title "Validation of rapid point-of-care diagnostic test for sexually transmitted infection for self-use among adolescent girls and young women". However, the authors are free to create another title.
  2. It is not clear why you mention the test to be used "among adolescent girls and young women in Johannesburg, South Africa". Could it be used in other cities/countries or among boys?
  3. The abstract represents the study properly.
  4. The introduction part does not provide a full rationale for the study. Please provide data on the local epidemiology of STIs, especially in the age group that you investigate (adolescents and young women). 
  5. What is the current testing systems used?
  6. Why do you suggest to test with rapid system only girls?
  7. What is the rationale to use the rapid system?
  8. In the methods section clearly describe the study subjects, inclusion and exclusion criteria.
  9. Results section - suggest rename the "3.1 Demographics" to a more academic sounding equivalent "3.1 Study participants' socio-demographic characteristics"
  10. In the discussion part the study strengths and limitations should be reported
  11. What is the clinical implication of the study results?
  12. What is the future research directions of the stuy results?

Reviewer 3 Report

Comments and Suggestions for Authors

Validation of rapid point-of-care sexually transmitted infection diagnostic tests for discrete self-testing use among adolescent girls and young women in Johannesburg, South Africa.

The study aimed to evaluate several commercial self-testing kits for sexually transmitted infectious diseases that were reported to have better sensitivity, specificity, PPV, and NPV. However, Table 1 shows these commercial kits' sensitivity, specificity, PPV, and NPV. But there are no bibliographic references. Who reported these sensitivities and specificities? How were these kits evaluated? How were they selected? Also, there are no comments about it in the discussion section.

The Materials and Methods section.

The Materials and Methods section contains no information on patient non-inclusion or exclusion criteria. Why? Were the use of antibiotics for other reasons, such as respiratory or digestive infections, permitted, or were samples excluded?

Endocervical cells, obtained from a thorough endocervical scraping, are required to diagnose Chlamydia infection. When the endocervix is infected, this tissue is friable and bleeds easily. Were these types of samples with blood accepted, or were there no problems? Was the patient able to perform the endocervical swab? Did they ensure that the swab had a good number of cells?

Self-sampling could be done at any time, before or after bathing, before having sex, or 24, 48, or 72 hours after engaging in this activity.

The results section is adequate. Despite the above, sensitivity, specificity, PPV, and NPV are needed when comparing two methods of diagnosis, but are not sufficient. To strengthen this information, it would help if you conducted the concordance and kappa analyses between the laboratory-collected and self-collected samples and the likelihood ratio.

Other questions and comments are: On many occasions, there can be great mucus in the vaginal. In the case of self-screening, the mucus was previously eliminated with a swab. How many swabs were provided to the patient, and what material were those?  

Also, there is little information on selecting patients because they are participating in another clinical study. However, it is essential to know why these tests are necessary. Were the patients with chronic infections due to gonorrhea, chlamydiosis, and trichomoniasis?

Were they treated with an antimicrobial before or during self-screening?

 It is important to note that a negative self-screening test does not necessarily mean that you do not have an infection despite showing some clinical signs of vaginitis, since the etiological agents causing this pathology could be several.

The discussion section. There is no discussion or comment on why several self-tests are negative compared to molecular screening, when these tests have been reported to have sensitivity and specificity of up to 90-100% and 96%-99%, respectively. There is no discussion with the experience of the other researcher about the advantages and disadvantages of self-test.

The reference section. For references 23-27, it would help if you put the web address. 

Round 2

Reviewer 1 Report

Comments and Suggestions for Authors

The article 'Validation of Rapid Point-of-Care Sexually Transmitted Infection Diagnostic Tests for Self-Test Use Among Adolescent Girls and Young Women' is a highly relevant study that evaluates the performance of rapid point-of-care STI diagnostic tests for self-testing among this demographic. The authors competently describe the methodology, which incorporates cross-sectional analyses and a longitudinal study. This dual approach effectively assesses test performance in clinical settings and in real-world home-use scenarios.
In response to reviewer comments, the authors addressed inaccuracies and typos and provided clarifications regarding the statistical methodology and refinements to the regional data. These improvements have enhanced the clarity of the presentation, making the results more transparent and justifiable.
Despite limitations regarding sampling in the longitudinal study and specific test kit usage protocols, the article demonstrates the significant potential of STI self-testing. Overall, the corrections have positively impacted the paper's quality, making it a valuable contribution to the development of practical solutions for improving STI screening and early diagnosis. I wish the authors every success.

Reviewer 2 Report

Comments and Suggestions for Authors

Thank you for addressing the comments. The manuscript was substantially improved.

However, the title is still not well-sounding. Please revise it.

Reviewer 3 Report

Comments and Suggestions for Authors

The manuscript requires several changes, many of which have been included in the manuscript.
